# The Impact of Phytase and Different Levels of Supplemental Amino Acid Complexed Minerals in Diets of Older Laying Hens

**DOI:** 10.3390/ani13233709

**Published:** 2023-11-29

**Authors:** Waleska R. L. Medeiros-Ventura, Carlos B. V. Rabello, Marcos J. B. Santos, Mércia R. Barros, Rogério V. Silva Junior, Heraldo B. Oliveira, Fabiano S. Costa, Andresa G. Faria, Alba K. Fireman

**Affiliations:** 1Animal Science Department, Universidade Federal Rural de Pernambuco, Rua Dom Manoel de Medeiros, s/n, Dois Irmãos, Recife 52171-900, Brazil; waleska_medeiros@hotmail.com (W.R.L.M.-V.); carlos.rabello@ufrpe.br (C.B.V.R.); rogerio-ventura@hotmail.com (R.V.S.J.); heraldoliveira1@gmail.com (H.B.O.); andresafaria2017@gmail.com (A.G.F.); 2Veterinary Science Department, Universidade Federal Rural de Pernambuco, Rua Dom Manoel de Medeiros, s/n, Dois Irmãos, Recife 52171-900, Brazil; mercia.barros@ufrpe.br (M.R.B.); fabianosellos@hotmail.com (F.S.C.); 3Zinpro Corporation, Eden Prairie, MN 55344, USA; afireman@zinpro.com

**Keywords:** egg production, enzyme, shell resistance, trace minerals

## Abstract

**Simple Summary:**

A study was conducted to investigate how different mineral supplements, along with an enzyme called phytase, impact egg-laying hens. This study involved 512 hens, divided into 8 groups, who were given 8 different diets. The main differences in the diets were: (1) the presence of the phytase enzyme; and (2) the proportion of minerals derived from amino acid-bound minerals. One group received 100% of the minerals from amino acid-bound sources; another group received 70%; and another group received 40%. Additionally, there were two control groups with and without phytase. The diet with 70% amino acid-bound minerals, combined with the phytase enzyme, resulted in the highest egg-laying percentage and the most efficient feed conversion. Eggs from the groups receiving 40%, 70%, and 100% amino acid-bound minerals had heavier yolks, while the group receiving only inorganic minerals had lighter yolks and whites. The groups receiving 100% and 70% amino acid-bound minerals produced eggs with the strongest eggshells. The ideal amounts of amino acid-bound minerals were: 42 mg kg^−1^ zinc, 49 mg kg^−1^ manganese, 5.6 mg kg^−1^ copper, 28 mg kg^−1^ iron, 0.175 mg kg^−1^ selenium, and 0.70 mg kg^−1^ iodine.

**Abstract:**

A study was conducted to evaluate the effects of different sources and levels of supplemental amino acid-complexed minerals (AACM), with and without enzyme phytase (EZ). A total of 512 Dekalb White laying hens at 67 weeks of age were used in a 2 × 3 + 2 factorial arrangement of 8 treatments and 8 replications each. The main effects included EZ supplementation (600 FTU kg^−1^) and AACM inclusion level (100%, 70%, and 40% of inorganic mineral recommendations), plus two control treatments. The group of hens fed AACM-100 showed lower feed intake than the inorganic mineral (IM) group. The diet containing AACM-EZ-70 provided a higher (*p* < 0.05) laying percentage and a lower (*p* < 0.05) feed conversion ratio than both the IM and IM-EZ diets. The groups fed AACM-EZ-40, AACM-EZ-100, and AACM-70 produced heavier yolks (*p* < 0.05). Hens fed IM laid eggs with the lowest yolk and albumen weights (*p* < 0.05). Layers fed with AACM-100 and AACM-70 produced the most resistant eggshells to breakage (*p* < 0.05). In diets containing phytase, the optimal AACM recommendations for better performance and egg quality in older laying hens are: 42, 49, 5.6, 28, 0.175, and 0.70 mg kg^−1^ for Zn, Mn, Cu, Fe, Se, and I, respectively.

## 1. Introduction

Aging in laying hens is associated with significant changes in egg quality, primarily in external characteristics. As the hen ages, eggshell thickness (ST) decreases, and its resistance to breakage worsens. Consequently, research has been conducted to identify alternatives for maintaining eggshell quality, especially for layers at an advanced age. However, most studies have primarily focused on the effects of macrominerals, such as calcium (Ca) and phosphorus (P). Hence, investigations are required to explore the supplementation of trace minerals, including zinc (Zn), manganese (Mn), copper (Cu), iron (Fe), selenium (Se), and iodine (I), since they play crucial roles as enzyme cofactors in animal metabolism [1,2]. Supplementing trace minerals in laying hens’ diets can improve enzymatic release, which leads to enhancements in the ultrastructure of eggshells [3,4], ultimately resulting in improved mechanical strength.

The industry typically links the use of inorganic sources of microminerals (IM) with the enzyme phytase in laying hen diets. Despite being relatively inexpensive, IM exhibit low bioavailability, as they dissociate into active cations upon reaching the gastrointestinal tract, where they interact with other dietary components such as phytic acid, folic acid, and tannins, forming insoluble complexes and rendering them unavailable to the animal [5]. Therefore, the use of phytase, coupled with these IM, exacerbates the formation of insoluble complexes and competition for absorption sites in the intestinal lumen, as its use stimulates the release of not only phytic P but also various trace minerals [6,7,8].

In this scenario, the utilization of minerals linked to organic molecules (OMM) is adopted as an alternative to minimize the problems associated with the use of IM in laying hen diets. This is because OMM can be absorbed at different locations in the gastrointestinal tract, thereby avoiding ionic competition [9]. Conversely, there are various types of OMMs, such as proteinates, glycinates, metal-amino acid chelates, and amino acid-complexed minerals (AACM), each exhibiting different absorption capacities [10,11].

In the digestive tract, AACM is highly stable and chemically inert, thus not interacting with free metal ions. Upon absorption, they pass directly into the plasma through the cells of the intestinal mucosa, which leaves their bond unchanged [9]. Due to this complex, the movement of minerals across the mucosa becomes mediated by amino acid transporters [12]. Therefore, not only their chemical form but also the level of supplementation in the diet require further study.

Stefanello et al. [13] reported that Mn, Zn, and Cu supplementation in proteinate form did not affect performance variables such as average daily feed intake (ADFI), feed conversion ratio (FCR), egg output (EO), or egg losses. However, it improved the eggshell quality traits, with the best results achieved at the Mn, Zn, and Cu supplementation levels of 65, 60, and 10 mg kg^−1^, respectively. Li et al. [14], on the other hand, worked with Zn sulfate and Zn methionine and found that a Zn methionine level of 80 mg kg^−1^ reduced ADFI and improved FCR. There was also a significant effect on egg quality, with the Zn methionine levels of 60, 80, and 100 mg kg^−1^ increasing egg weight (EW), albumin height (AH), and Haugh unit, respectively.

Although some studies have investigated the effects of trace minerals on egg performance and quality, none have considered the simultaneous supplementation of Zn, Cu, Mn, Fe, I, and Se in the form of AACM for laying hens. Given the diverse results found in the literature and the limited research on the simultaneous use of trace minerals, further investigation is needed to evaluate the combined use of these micronutrients at varying levels.

Therefore, this study was conducted to examine the effects of supplementation with differing sources and levels of trace minerals, with or without the addition of phytase, on the performance, egg quality, hormonal concentration, hematological profile, and organ weight of older laying hens. The hypothesis tested in this study is that the use of AACM at low levels can maintain or improve performance and egg quality without compromising the hematological or hormonal variables of the birds.

## 2. Materials and Methods

### 2.1. Animal Ethics 

This research project was previously approved by the Animal Research Ethics Committee (CEUA) at the Federal Rural University of Pernambuco (approval no. 95/2018).

### 2.2. Experimental Site and Animal Management

An experiment was conducted at the Federal Rural University of Pernambuco in the Small-Animal Experimental Unit of Carpina, Pernambuco, Brazil. A total of 512 Dekalb White laying hens at 68 weeks of age were distributed into 64 experimental cages equipped with trough-type feeders and automatic drinkers with attached cups. The experimental period consisted of 5 cycles of 28 days each, totaling 140 days. During this period, water was available *ad libitum*, whereas the feed was adjusted as recommended by the Dekalb White guideline. The lighting program adopted consisted of 17 h of light per day (12 h natural light + 5 h artificial light). 

### 2.3. Experimental Design and Diets

A total of 512 Dekalb White laying hens were distributed by weight and EO in a completely randomized experimental design with a 2 × 3 + 2 factorial arrangement with 8 treatments, 8 replicates, and 8 birds per treatment. The first factor referred to diets with amino acid-complexed minerals (Zn, Mn, Cu, Fe, I, and Se), without (AACM) or with (AACM-EZ) the addition of 600 FTU kg^−1^ phytase. The second factor corresponded to 3 levels of AACM inclusion (100%, 70%, or 40% AACM). Inclusion levels were based on the IM requirements of the Dekalb White manual guideline, which are 60, 70, 8, 40, 1.0, and 0.250 mg kg^−1^ of Zn, Mn, Cu, Fe, I, and Se, respectively (Table 1). Two control treatments were used, which corresponded to IM supplementation with (IM-EZ) or without 600 FTU kg^−1^ phytase.

Trace minerals included were acquired from Zinpro^®^ Performance Minerals^®^. The trace microminerals Zn, Mn, Cu, and Fe were complexed using a non-specific amino acid ligand. Supplemental iodine was associated with the Zn molecule, while Se was supplied as Zn-L-selenomethionine. Experimental treatments included 2 basal diets (with and without the addition of phytase), where mineral premix was modified to create each treatment group. 

### 2.4. Mineral Concentrations in Feed and Water

Feed samples were collected, placed in plastic bags, and stored in a freezer at −20 °C until analysis. Subsequently, feed samples were ground using a ball mill and dried in an oven at 105 °C. The resulting samples, weighing 0.5 g, were digested in 6 mL of nitric acid and diluted to 25 mL with deionized water. Wet digestion of the samples was performed using a microwave for 30 min. Mineral quantification was carried out using inductively coupled plasma optical emission spectroscopy (ICP-OES). During the experimental period, water samples were collected in plastic containers and frozen until analysis. Mineral quantification in water was also carried out using an inductively coupled plasma source (Optima 7000 DV ICP-OES, PerkinElmer, Greenville, SC, USA). The trace minerals from the diets and premix compositions are presented in Table 2.

### 2.5. Performance Variables

The following performance variables were evaluated: EW (g), EO (%), egg mass (EM, g bird^−1^ day^−1^), ADFI (g bird^−1^ day^−1^), and FCR (kg kg^−1^). Eggs were collected twice daily, counted, and weighed. The EO was calculated as the ratio between the number of eggs produced and the number of birds housed. The EM was calculated by multiplying EO by EW. The FI was measured by subtracting the amount of feed left over from the amount of feed supplied over a seven-day period and then dividing this result by the number of birds housed per experimental unit. The FCR was calculated as the ADFI divided by the EM determined in the same period. 

### 2.6. Egg Quality

During the last 3 days of each experimental period, 3 eggs were selected per plot based on their average weight, resulting in a total of 24 eggs per treatment. These eggs were then used to measure various quality variables, including EW (g), albumen weight (AW, g), yolk weight (YW, g), shell weight (SW, g), AH (mm), shell thickness (ST, mm), yolk color (YC), and Haugh unit (HU).

Each egg was identified and individually weighed on a semi-analytical scale with 0.01 g precision (Bel, L 3102iH). To determine AH, the eggs were broken, and the contents (albumen + yolk) were placed on a flat, level surface. The AH (mm) was then measured by reading the value indicated on the digital caliper (Absolute Digital AOS, Mitutoyo, SP, Brazil; 0.01 mm precision). The HU was calculated using EW (g) and AH (mm), which were entered into the following equation: HU = 100 log(AH + 7.57 − 1.7 EW^0.37^) [15].

Subsequently, the yolk was separated and weighed on a semi-analytical scale with 0.01 g precision. Eggshells were washed in running water shortly after breaking to remove all albumen residue, and then air-dried for 48 h before being weighed. After weighing, a precision micrometer (iGaging, San Clemente, CA, USA) was used to measure the thickness of 3 eggshell regions (basal, equatorial, and apical), from which the average thickness was calculated per egg. The AW was determined by subtracting the sum of the SW and YW from the EW. The YC was evaluated using a colorimetric fan with a scale between 1 and 15.

### 2.7. Organ Weight

On the last day of the experimental period, 1 bird was selected per plot according to the average weight of each experimental unit to be euthanized for organ collection. After euthanasia by cervical dislocation, the following organs were collected and weighed on a semi-analytical scale (±0.01 g): spleen, liver, pancreas, intestine, and oviduct. Additionally, intestinal length was also measured.

### 2.8. Collection and Preparation of Samples for Mineral Quantification 

#### 2.8.1. Egg Yolk

At the end of the last cycle, 3 yolks were collected per experimental unit, stored in plastic bags, and frozen at −20 °C. Afterwards, they were thawed and homogenized. At the Animal Nutrition Laboratory of UFRPE (LNA/UFRPE), a 10 g sample of yolk from each replicate was dried in a forced-air oven at 105 °C for 24 h.

#### 2.8.2. Tibia 

At the end of the experimental period, 1 bird was selected per experimental plot according to the average weight of each plot and euthanized by cervical dislocation to collect tibia. Left tibias were collected and stored in labeled plastic bags. Tibia samples were frozen at −20 °C until analysis. The tibias were sent to LNA/UFRPE, where they were thawed and dried in a forced-air oven at 105 °C for 24 h. Tibias were then placed in crucibles and calcined in a muffle furnace (model 2000F; Zezimaq, Minas Gerais, Brazil) at 600 °C for 4 h to obtain ash values.

#### 2.8.3. Digestion and Quantification of Minerals in Samples 

After obtaining the dry samples of yolk, 0.5 g of each sample was weighed on an analytical scale (±0.0001 g) and digested with 6 mL of HNO_3_ (65%) in a microwave oven (Mars Xpress: Technology Inside, CEM Corporation) for 30 min at 160 °C. The obtained solution was filtered through quantitative blue ribbon filter paper and diluted with deionized water until reaching a volume of 25 mL. For the tibias, 0.5 g samples were weighed and digested in an open system with 6 mL of HNO_3_ (65% AR) for 10 min. After this interval, the samples were subjected to the same filtering and dilution process mentioned above for the other samples. Mineral Quantification (Zn, Mn, Cu, Fe, Ca, and P) in the samples was performed at the Environmental Soil Chemistry Laboratory at UFRPE using an optical emission spectrophotometer with an inductively coupled plasma source (Optima 7000 DV ICP-OES, PerkinElmer).

### 2.9. Computerized Densitometry

Computed tomography was performed on 5 tibias per treatment using a Hi-Speed FXI CT scanner (General Electric, Fairfield, CT 06824, USA) in the FOCUS laboratory. The tibias were placed side by side for image acquisition. Cross-sectional images were acquired from 2 mm thick sections with a reconstruction interval of 1 mm at 120 kV and automatic tube current (mA) at the speed of one rotation per second.

Subsequently, the images were analyzed using Dicom software (version 1.1.7, Horos, Purview, Annapolis, MD, USA) to estimate the individual values of bone radiodensity at 3 different diaphysis section levels (proximal, medial, and distal). Each region was divided into 4 quadrants, and a circular region of interest was selected for densitometric evaluation of the cortical bone. Results were expressed in Hounsfield units (HOU), which were subsequently corrected and converted to mg/cm^3^ of Ca hydroxyapatite by the equation BMD = 200 HOUt/(HOUw − HOUb), following the methodology described by [16] and [17].

### 2.10. Hematological Profile

The hematological profile was analyzed by randomly selecting one bird per replication in the last experimental week and harvesting 4 mL of blood by jugular venipuncture into a heparin-containing tube. The collected blood samples were then sent to the veterinary laboratory, LaborVet. Red blood cells, leukocytes, and platelets were counted using a Newbauer chamber after dilution with Natt–Herrick reagent, and the count was performed under a microscope (Olympus America CX-41, Center Valley, PA 18034-0610, USA). Hematocrit was determined using the microcapillary method, and total plasma protein was measured by refractometry. Differential leukocyte counts were performed by reading the slide under an optical microscope (Olympus America CX-41, Center Valley, PA 18034-0610, USA) with staining using the fast panoptic method.

Blood samples were also collected in the last week from 1 bird per experimental unit for hormonal and biochemical analysis. These samples were collected and centrifuged (ELMI Centrifuge CM-MT) at 3500 rpm for 15 min. Using a pipette, 2 mL of serum was collected, stored in Eppendorf tubes, and frozen at −20 °C until analysis. The samples were then sent to the Laboratory of Molecular Biology Applied to Animal Production (BIOPA/DZ/UFRPE) for analysis of hormonal concentrations of corticosterone (μg mL^−1^), triiodothyronine (T3) (μg mL^−1^), and alkaline phosphatase (U L^−1^). At the time of analysis, the samples were thawed at room temperature, diluted, and prepared according to the methodology described by the commercial kit (BIOCLIN^®^) and subsequently read in a spectrophotometer (Bioclin, Biolisa Reader).

### 2.11. Statistical Analysis

The normality and homoscedasticity assumptions were tested for analysis of variance. Data were analyzed using the PROC GLM procedure of Statistical Analysis System software version 9.4 [18]. In the case of statistical differences between the AACM diets, the means were compared by Tukey’s test (*p* < 0.05). To compare the inorganic treatments (IM and IM-EZ) with the AACM source, Dunnett’s test (*p* < 0.05) was applied.

The statistical model is:Y_ij_ = μ + α_i_ + β_j_ + (αβ)_ij_ + ε_ij_(1)
where:

Y_ij_ is the response variable (e.g., egg production, feed intake, etc.) for the i-th level of the first factor and the j-th level of the second factor.

μ is the overall mean.

α_i_ is the effect of the i-th level of the first factor (diets with AACM without or with phytase).

β_j_ is the effect of the j-th level of the second factor (100, 70, or 40% AACM).

(αβ)_ij_ is the interaction effect between the i-th level of the first factor and the j-th level of the second factor.

ε_ij_ is the random error term for the i-th level of the first factor and the j-th level of the second factor.

## 3. Results

### 3.1. Performance

Supplementation with AACM did not result in a significant effect on EW (*p* = 0.36) or EM (*p* = 0.10), as shown in Table 3. However, a significant interaction was observed for EO (*p* = 0.02), ADFI (*p* = 0.36), and FCR (*p* = 0.03). Birds supplemented with AACM-EZ-70 showed a 2.3% increase in EO compared to those fed with AACM-70. Among the treatments, laying hens supplemented with AACM-100 showed the lowest ADFI. Similarly, AACM-EZ-100 resulted in 7.1 g of ADFI to produce 1 kg of eggs compared to AACM-70. Dunnett’s test (Figure 1) showed a significant effect on EO (*p* < 0.01), ADFI (*p* = 0.03), and FCR (*p* = 0.01). Supplementation with AACM-EZ-70 resulted in higher EO and EM and a slight improvement in FCR compared to the IM diets (IM and IM-EZ). The AACM-40 diet also provided a better FCR than the IM-EZ treatment.

### 3.2. Egg Quality

The diet did not have a significant impact on egg quality for EW (*p* = 0.68), SW (*p* = 0.16), AW (*p* = 0.68), AH (*p* = 0.58), ST (*p* = 0.14), and HU (*p* = 0.60), as shown in Table 4. However, an interaction effect was observed for YW (*p* = 0.01), with the AACM-EZ-40 treatment resulting in yolks that were 2.3% heavier than those from hens fed AACM-100. When the factors were evaluated separately, it was found that supplementation with AACM-100 and AACM-70, as well as the addition of phytase to the diets, intensified the YC. Birds fed IM or IM-EZ diets were not statistically different (*p* > 0.05) from the other treatments for EW, YW, SW, AH, and Haugh units (Figure 2). The IM group also demonstrated a lower intensity of yolk color when compared to the AACM-EZ-100 and AACM-EZ-70 groups (*p* < 0.05). Moreover, the AW in the IM group was lower in comparison to the others but did not differ from the AACM-EZ-40 group (*p* < 0.05). Additionally, layers supplemented with AACM-100, AACM-EZ-70, and AACM-EZ-40 exhibited an increase in ST (*p* < 0.05) compared to those fed with IM and IM-EZ diets.

### 3.3. Organ Weight

There was a 10.7% increase in pancreas weight (*p* = 0.02) for birds fed AACM without phytase in their diets. However, the liver (*p* = 0.99), spleen (*p* = 0.47), intestines (*p* = 0.40), and oviduct (*p* = 0.58) were not influenced by phytase or mineral supplementation level (Table 5). Furthermore, no significant difference (*p* > 0.05) in organ weight variables was observed between birds that consumed the IM diet and those that were fed diets with AACM, as shown in Figure 3.

### 3.4. Egg Yolk Mineral Deposition

The treatments had no significant effect on the deposition of Cu (*p* = 0.88), Mn (*p* = 0.13), Fe (*p* = 0.67), Ca (*p* = 0.36), and *P* (*p* = 0.58) in the egg yolks of the hens (Table 6). However, the supplementation of phytase led to an increase in the deposition of Zn by 6.8% (*p* = 0.03). Moreover, a higher Se content in the egg yolks was observed with an increase in the level of AACM (*p* < 0.01). Birds fed IM and IM-EZ had higher levels of Se than AACM-40 and AACM-EZ-40, and the IM-EZ group had a higher deposition of Zn than those fed AACM-70 (*p* < 0.05; Figure 4).

### 3.5. Tibia Mineral Deposition

Treatments had no significant effect on the deposition of Zn (*p* = 0.15), Cu (*p* = 0.47), Mn (*p* = 0.14), Fe (*p* = 0.68), Ca (*p* = 0.85), and P (*p* = 0.87) in the tibias of hens (Table 6). Birds fed AACM-EZ had a higher concentration of Mn in the tibia than the group fed AACM (*p* = 0.01). The results of this study revealed that hens fed IM-EZ exhibited significantly higher levels of Mn deposition in their tibia (Figure 5) compared to those fed with IM, AACM-100, and AACM-40 (*p* = 0.01). However, hens fed with IM had significantly lower levels of Mn in their tibia compared to those fed with AACM-EZ-100. Additionally, the Zn deposition in treatments AACM-100, AACM-70, and AACM-EZ-70 was higher than that in the IM-EZ group (*p* < 0.05).

### 3.6. Tibia Densitometry

In Table 7, the means of bone densitometry are shown. There was an interaction between diets and AACM levels on bone density in the 3 studied regions: proximal (*p* = 0.02), medial (*p* = 0.01), and distal (*p* = 0.04). The proximal tibia segment was 38% denser in birds supplemented with AACM-70 compared to those fed AACM-EZ-70. The bone density in the medial segment was highest in birds supplemented with AACM-EZ-100, while the lowest density was observed in birds fed AACM-100 diets. Similarly, birds fed AACM-100 showed lower bone density compared to other treatments. The density of the proximal tibia in birds from the IM treatment did not differ significantly from those fed AACM-100, but it was lower compared to all other treatments (*p* < 0.05; Figure 6). The treatments AACM-70 and AACM-40 had a higher density compared to IM-EZ, but there was no significant difference between these treatments (*p* < 0.05). In the medial portion of the tibia, IM had a lower density than AACM-70, IM-EZ, and all AACM-EZ treatments, regardless of the supplementation level (*p* < 0.05). The IM-EZ group had a higher density compared to AACM-100 and IM, but not to the other treatments. In the cortical segment of the tibia, laying hens fed IM had a lower density than the other treatments but not compared to AACM-100, and IM-EZ had a lower density than AACM-EZ-70 and AACM-EZ40 (*p* < 0.05).

### 3.7. Hematological Profile

Supplementation of laying hen diets with different levels of AACM did not affect the hematological profile of the hens (*p* > 0.05). However, supplementation with phytase increased the concentration of red blood cells (*p* = 0.01) in the plasma (Table 8). Birds fed IM-EZ had lower concentrations of red blood cells than those laying hens fed AACM-EZ-100 and AACM-EZ-40 (*p* < 0.05); however, there were no differences in the other blood variables (Figure 7). Furthermore, supplementation with AACM did not cause any significant differences (*p* > 0.05) in the hormonal concentrations of T3 (μg mL^−1^), corticosterone (μg mL^−1^), or alkaline phosphatase (U L^−1^). The Dunnett test also showed no effect on these variables (*p* > 0.05; Figure 8).

## 4. Discussion

This study demonstrates that supplementing laying hen diets with a combination of AACM and phytase enzyme can improve their performance, egg quality, and pancreas weight. Additionally, the supplementation of AACM was found to increase the synthesis of red blood cells in older hens. These results suggest that AACM supplementation may have practical implications for improving performance efficiency and egg quality in laying hens. 

Previous studies have refuted the hypothesis that trace mineral supplementation influences laying hen performance [13,19,20,21]. However, our study found that AACM supplementation had a favorable impact on laying and feed efficiency. The discrepancy in outcomes between our study and previous research could be attributed to the differential nature of the supplemented molecules. There are different organic molecules bound to trace minerals, and they exhibit distinct characteristics and variations in bioavailability between them [22]. For example, 2-hydroxy-4-methylthiobutyric acid (HMTBa) chelated to trace minerals is bound to 2 molecules of HMTBa, making it a large and unstable molecule that cannot be absorbed without dissociating at low pH. Proteinates are minerals bound to large protein molecules, and they are susceptible to digestion in the intestine. Additionally, at low pH, the protein-bound mineral may dissociate, resulting in reduced absorption and utilization of the mineral.

Our study also found that Mn and Zn likely contributed to the improvement in laying hen performance. Manganese acts as a cofactor in regulating cholesterol synthesis, which acts as a precursor to steroid hormones such as estrogen and progesterone. Additionally, Mn can stimulate the release of luteinizing hormone, a hormone that triggers ovulation and influences production performance in laying hens [23]. Zinc may also play a role, as plasma Zn concentration can serve as an indicator of vitellogenin, a yolk precursor [24]. Our findings suggest that the supplementation of AACM can enhance plasma levels of vitellogenin in laying hens, potentially boosting EO.

The higher EO in the diet with 70% AACM may be attributed to the synergistic effect of minerals complexed with amino acids and those released by phytase. Phytase is an enzyme that breaks down phytic acid, an anti-nutrient that binds to minerals and reduces their bioavailability [25]. By breaking down phytic acid, phytase can release minerals that are available for absorption by the animal. Phytase’s ability to enhance P and mineral availability may have influenced the utilization of AACM components differently at this specific concentration, modulating metabolic processes and influencing EO [26]. Variations in enzymatic activity, nutrient absorption rates, and metabolic pathways could all contribute to the observed differences. The observed effects on FI and FCR may be attributed to the contribution of Zn, which acts through the mechanisms of increasing cell proliferation and suppressing apoptosis. This mechanism leads to improved digestive capacity and absorptive function of the gastrointestinal tract, resulting in more efficient utilization of nutrients and ultimately better production performance [25].

The birds supplemented with IM, in general, showed lower results. This may be due to the ionization of IM in the stomach, which can result in losses during the process of being captured by intestinal cells through passive diffusion or active transport. These losses may occur due to reactions with other nutrients, rendering them insoluble or competing with different minerals to bind to the carriers. Gao et al. [26] observed a decrease in intracellular Cu^++^ levels when AACM was compared to free IM forms, suggesting that AACM has a higher absorption rate. Medeiros-Ventura et al. [27] have shown that replacing IM with AACM (Zn, Mn, and Cu) improved the performance of commercial laying chicks during the starter phase, resulting in better weight gain and FCR. Pereira et al. [28] also observed that birds supplemented with AACM from the starter phase reached 35% production 2 days earlier than those fed IM, which is attributed to the superior development of the oviduct, leading to better physiological growth in the birds. Similarly, [14] found that EO increased with supplementation of 80 and 60 mg kg^−1^ of Zn methionine. 

The production of high-quality eggs has generated continued interest, leading to research focusing on minerals used in the nutrition of commercial laying hens. In this study, supplementation with AACM influenced both YW and AW, resulting in significantly heavier yolks and albumen weight. The development and functioning of the avian reproductive system rely on ovarian hormones, which stimulate the synthesis of egg yolk proteins [29] and can be influenced by dietary Mn [30]. Gonadal steroid hormones are also involved in inducing synthesis by the tubular glands and epithelial cells of the magnum [31]. Supplementation with organic minerals can affect morphometric variables of the reproductive tract, particularly the magnum, by reducing oxidation and enhancing cell integrity, resulting in improved oviduct morphology and increased numbers of albumen-secreting cells [32].

This investigation demonstrated that YC is influenced by the presence of phytase in diets and different levels of AACM supplementation. The observed color intensity may be attributed to the ability of phytic acid to form insoluble complexes with pigment molecules. Incorporating phytase in the diets of laying hens may catalyze the hydrolysis of phytic acid, thereby increasing pigment absorption and intensifying YC [33]. Free carotenoids are absorbed through micelles alongside fatty acids and transported in the bloodstream via lipoproteins [34]. Their molecular structure contains conjugated double bonds, which render them susceptible to oxidation, necessitating the addition of stabilizing agents to maintain color [35]. 

Eggshell quality is a crucial aspect of the poultry industry’s economic viability, as losses due to egg breakage can account for 8% to 10% of total production [36]. In this study, supplementation with AACM increased ST when compared to IM diets. The composition of the eggshell comprises organic and inorganic components, with the matrix, eggshell membrane, mammillary knob, and cuticle being notable organic components [37], and Ca carbonate crystals being the primary inorganic constituents [38]. Therefore, the factors that influence eggshell thickness are complex and reflect the intricate interplay between these elements. 

The trace minerals Mn, Zn, and Cu have been shown to have a direct impact on eggshell quality due to their catalytic properties as key enzymes involved in the membrane formation process or through a modifying effect on the calcite crystal growth mechanisms during shell formation [39]. Some studies have indicated that the thickness and strength of the eggshell are influenced by the relationship between the palisade and mammillary layers, the density and width of the mammillary knobs, and the organization of calcite crystals, which are also affected by dietary mineral levels [2,13,40,41].

Manganese, for instance, acts as an activator of the glycosyltransferase enzyme, which is involved in the synthesis of glycosaminoglycans and glycoproteins, thus contributing to the development of the organic matrix by forming nucleation sites and regulating the growth and orientation of calcite crystals during eggshell formation [41]. Therefore, the proteoglycan components of the organic matrix play a critical role in influencing eggshell structure and breaking strength [42]. Zinc is another element that affects eggshell quality, as it is a component of several metalloenzymes, including carbonic anhydrase, which facilitates the conversion of carbon dioxide into bicarbonate and, consequently, regulates the transfer of bicarbonate ions from the blood to the eggshell gland during egg formation [43,44]. Inadequate dietary Cu levels result in malformation of the eggshell, characterized by an abnormal distribution of the shell membrane fibers due to changes in lysine-derived cross-links, as Cu is a component of the lysyl oxidase enzyme involved in the conversion of lysine into desmosine cross-links and isodesmosine [45].

Eggs are an important component of the human diet, particularly in impoverished countries where other protein sources may not be readily available. Eggs with higher concentrations of trace minerals are significant in preventing diseases such as cancer, heart disease, and osteoporosis [46]. The reduction of reactive oxygen species is one of the chemical reactions involved in these processes. In our findings, we observed higher deposition of Se and Zn in egg yolks, in which both molecules are involved in the synthesis and regulation of reactive oxygen species. However, diets supplemented with phytase influenced Zn deposition but not Se, where Se deposition was influenced by the levels of supplementation. The higher Zn levels in yolks may be attributed to the association with chelated trace minerals and those released from diets through the phytase enzyme. Additionally, there was no supplementation of IM premix.

A study developed by [47] demonstrated that changes in Zn concentrations in the tibia are influenced not only by the source (Zn methionine and Zn sulfate) but mainly by the concentrations used in the diet, with a maximum deposition occurring regardless of the mineral source used. In our studies, we detected changes in the Mn concentrations in the tibia of the birds supplemented with AACM-EZ. This may be related to the same mechanism used by Zn in the egg yolk. In addition, it is possible that the levels of other minerals used in our studies were utilized in other biochemical processes and body tissues, and consequently, the stored proportion was not sufficient to increase the levels of the elements in the tibia.

The higher bone density in the 3 segments (proximal, medial, and distal) shows that adequate bone formation is also dependent on trace minerals, which have an important role in bone development. Throughout the lifespan of birds, bone tissue undergoes continuous remodeling, involving both formation and resorption. This dynamic process is essential for maintaining bone homeostasis, where osteoclasts break down bone tissue and osteoblasts synthesize the same tissue [48]. Iron is an important trace mineral that plays a pivotal role in bone metabolism by modulating both formation and resorption. Inadequate intake of Fe, either in excess or deficiency, can disrupt this complex balance of bone homeostasis, leading to enhanced osteoclast activity and consequent bone loss. Therefore, it is crucial to ensure adequate levels of dietary Fe are fed to laying hens for optimal bone health and EO [49]. Our results show that the presence of phytase at 70% of the AACM level decreased the cortical bone density; however, the higher EO may have depleted the Ca reserve in this segment to form the egg structure and deposit it in the yolk and albumen, as cortical storage Ca is the first to be removed for eggshell synthesis.

In the present study, it was observed that the inclusion of phytase in AACM diets resulted in a reduction of pancreas weight, which may be attributed to the interaction between phytic acid and digestive enzymes. Phytate was found to bind to endogenous proteases such as trypsin and chymotrypsin, as well as their precursors, during their passage through the digestive tract [50]. At certain concentrations of phytate, a ternary protein-metal-phytate complex can be formed, as both enzymes and their precursors have an affinity for Ca, further reducing the bioavailability of proteins and minerals [51]. Due to the formation of these complexes with digestive enzymes, the pancreas attempts to compensate for low enzyme levels by increasing the secretion of digestive proteases in the intestine in response to negative feedback mechanisms [52]. However, this can lead to long-term cellular hyperplasia. 

As part of this research, the use of phytase resulted in an increase in the concentration of red blood cells. Phytic acid, present in feedstuffs under natural conditions, has a high potential for complexation with positively charged molecules, such as Fe^+3^ and Cu^+2^ cations, forming insoluble complexes with phytate when in the ionic form [53,54]. Iron and Cu are essential minerals for the formation and maintenance of the integrity of hemoglobin and other blood proteins, including erythrocuprein, found in red blood cells and involved in oxygen metabolism [55,56]. The inclusion of phytase in the diet of laying hens allowed the release of Fe and Cu that were trapped in the phytate molecule, thus contributing to the increase in the concentration of red blood cells.

In this study, the blood levels of T3, alkaline phosphatase, and corticosterone were not influenced. Pereira et al. [28] observed similar results when working with the partial replacement of IM minerals with AACM, as they did not find significant differences in T3 levels. The hormone T3 is one of the thyroid hormones that acts by participating in the control of metabolic processes [57], and its synthesis is dependent on the micronutrients Zn, Se, and I [58]. The absence of significant responses in some variables in this study confirms and supports the replacement of IM by lower levels of AACM, considering that different levels of replacement were evaluated without harming the performance of laying hens with advanced age.

The results of this study provide support that supplementation with AACM was more effective than IM sources in improving the performance and egg quality of laying hens, as demonstrated by the thicker and more resistant shells. The addition of the phytase enzyme to diets resulted in a reduction in pancreas weight and an increase in red blood cell concentration in plasma.

## 5. Conclusions

In diets containing phytase, the optimal AACM recommendations to achieve maximum performance and egg quality in older laying hens are 42, 49, 5.6, 28, 0.175, and 0.70 mg kg^−1^ for Zn, Mn, Cu, Fe, Se, and I, respectively. In diets without phytase, the optimal AACM recommendation in old laying hens is 24, 28, 3.2, 16, 0.1, and 0.4 mg kg^−1^ for Zn, Mn, Cu, Fe, Se, and I, respectively.

## Figures and Tables

**Figure 1 animals-13-03709-f001:**
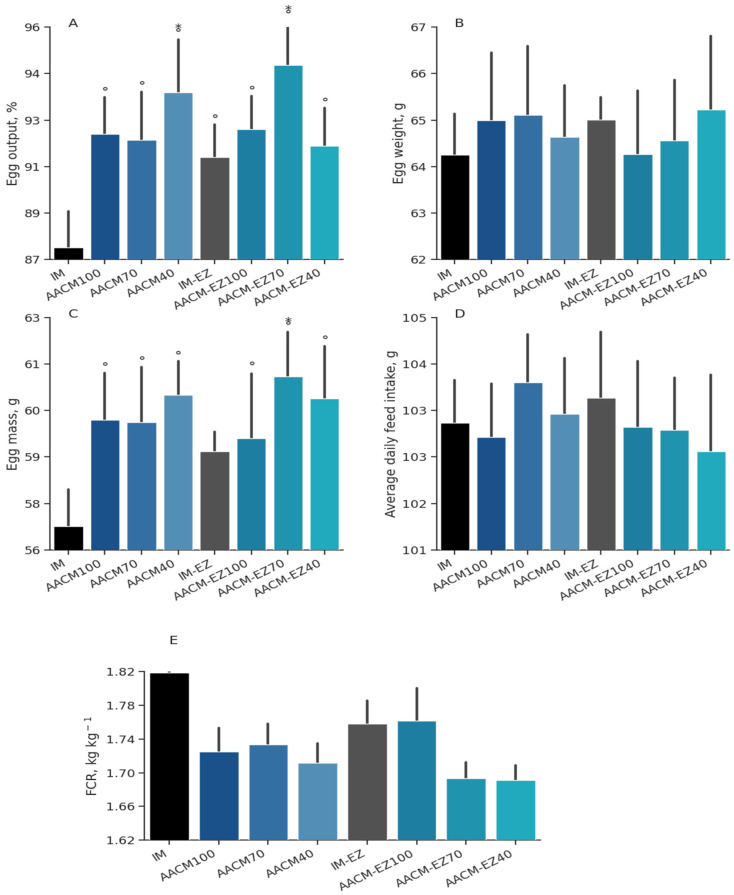
Egg output (**A**), egg weight (**B**), egg mass (**C**), average daily feed intake (**D**), feed conversion ratio (**E**) of laying hens from 68 to 90 weeks fed inorganic minerals (IM) and amino acid complex minerals (AACM) supplemented with or without the use of phytase (EZ). IM or IM-EZ: 60, 70, 8, 40, 1.0, and 0.250 mg kg^−1^ of Zn, Mn, Cu, Fe, I, and Se, respectively; AACM100 or AACM-EZ100: 60, 70, 8, 40, 1.0, and 0.250 mg kg^−1^ of Zn, Mn, Cu, Fe, I, and Se, respectively; AACM70 or AACM-EZ70: 42, 49, 6, 28, 0.700, and 0.175 mg kg^−1^ of Zn, Mn, Cu, Fe, I, and Se, respectively; AACM40 or AACM-EZ40: 24, 28, 3, 16, 0.400, and 0.10 mg kg^−1^ of Zn, Mn, Cu, Fe, I, and Se, respectively. Data were analyzed by Dunnett’s test (*p* < 0.05); ° Differs from IM treatment; * Differs from IM-EZ treatment.

**Figure 2 animals-13-03709-f002:**
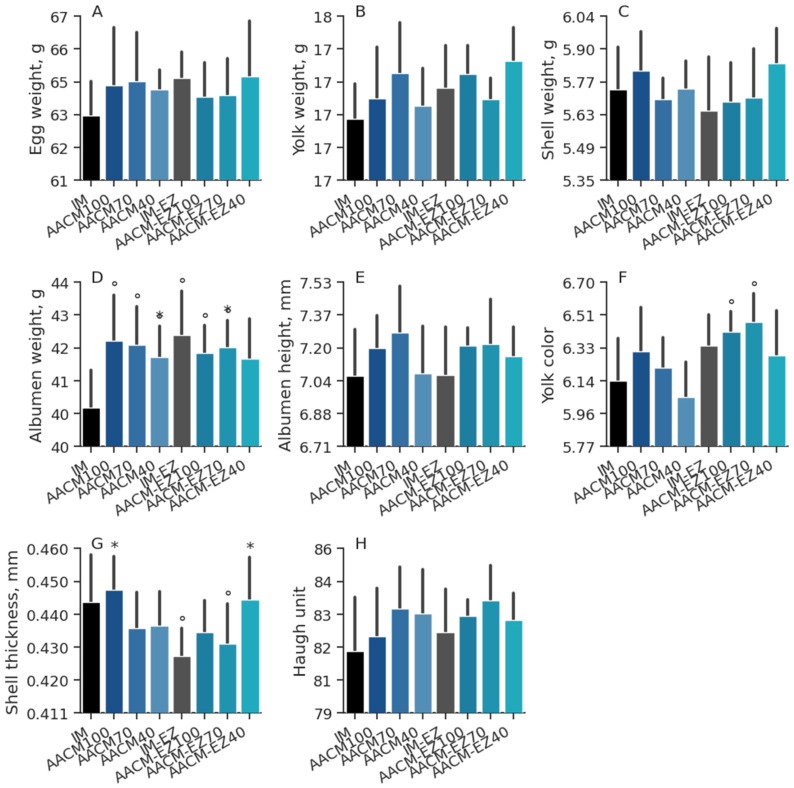
Egg weight (**A**), yolk weight (**B**), shell weight (**C**), albumen weight (**D**), albumen height (**E**), yolk color (**F**), shell thickness (**G**), and Haugh unit (**H**) of laying hens from 68 to 90 weeks fed inorganic minerals (IM) and amino acid complex minerals (AACM) supplemented with or without the use of phytase (EZ). IM or IM-EZ: 60, 70, 8, 40, 1.0, and 0.250 mg kg^−1^ of Zn, Mn, Cu, Fe, I, and Se, respectively; AACM100 or AACM-EZ100: 60, 70, 8, 40, 1.0, and 0.250 mg kg^−1^ of Zn, Mn, Cu, Fe, I, and Se, respectively; AACM70 or AACM-EZ70: 42, 49, 6, 28, 0.700, and 0.175 mg kg^−1^ of Zn, Mn, Cu, Fe, I, and Se, respectively; AACM40 or AACM-EZ40: 24, 28, 3, 16, 0.400, and 0.10 mg kg^−1^ of Zn, Mn, Cu, Fe, I, and Se, respectively. Data were analyzed by Dunnett’s test (*p* < 0.05); ° Differs from IM treatment; * Differs from IM-EZ treatment.

**Figure 3 animals-13-03709-f003:**
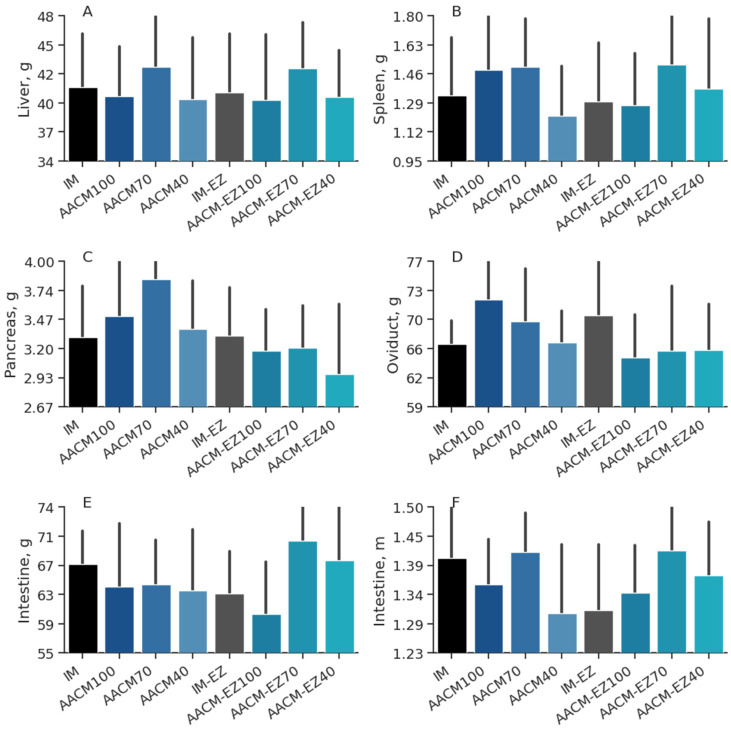
Liver (**A**), spleen (**B**), pancreas (**C**), oviduct (**D**), and intestine (**E**,**F**) of variables of laying hens from 68 to 90 weeks fed inorganic minerals (IM) and amino acid complex minerals (AACM) supplemented with or without the use of phytase (EZ). IM or IM-EZ: 60, 70, 8, 40, 1.0, and 0.250 mg kg^−1^ of Zn, Mn, Cu, Fe, I, and Se, respectively; AACM100 or AACM-EZ100: 60, 70, 8, 40, 1.0, and 0.250 mg kg^−1^ of Zn, Mn, Cu, Fe, I, and Se, respectively; AACM70 or AACM-EZ70: 42, 49, 6, 28, 0.700, and 0.175 mg kg^−1^ of Zn, Mn, Cu, Fe, I, and Se, respectively; AACM40 or AACM-EZ40: 24, 28, 3, 16, 0.400, and 0.10 mg kg^−1^ of Zn, Mn, Cu, Fe, I, and Se, respectively. Data were analyzed by Dunnett’s test (*p* < 0.05).

**Figure 4 animals-13-03709-f004:**
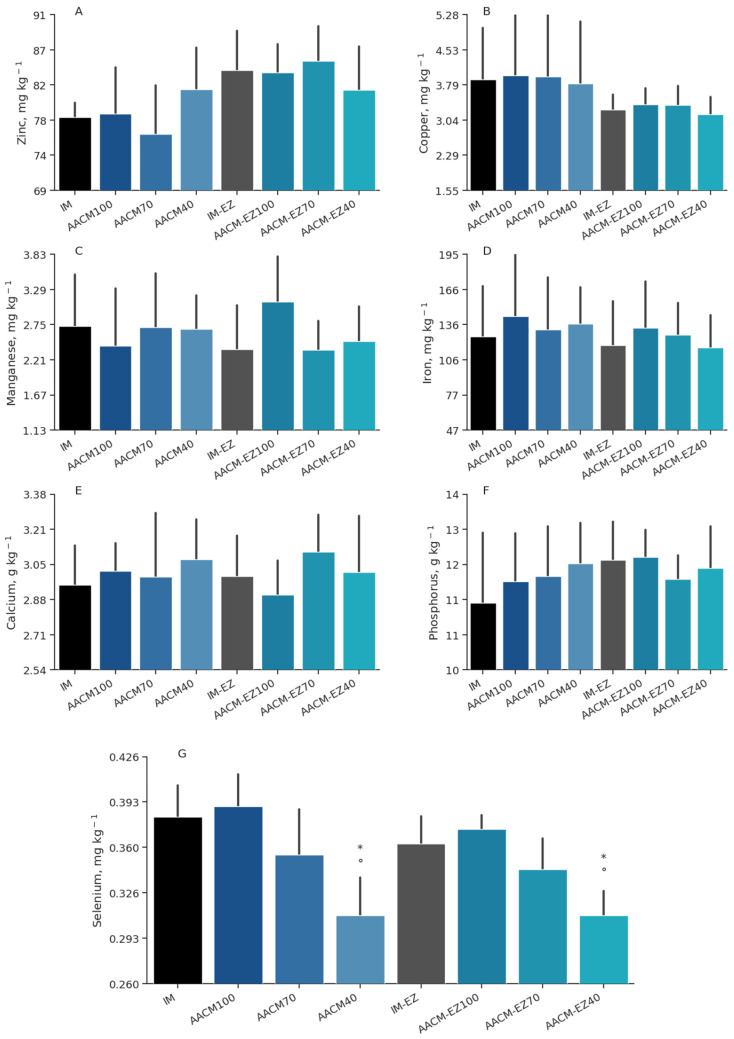
Zinc (**A**), copper (**B**), manganese (**C**), iron (**D**), calcium (**E**), phosphorus (**F**), and selenium (**G**) deposition in egg yolk of laying hens from 68 to 90 weeks fed inorganic minerals (IM) and amino acid complex minerals (AACM) supplemented with or without the use of phytase (EZ). IM or IM-EZ: 60, 70, 8, 40, 1.0, and 0.250 mg kg^−1^ of Zn, Mn, Cu, Fe, I, and Se, respectively; AACM100 or AACM-EZ100: 60, 70, 8, 40, 1.0, and 0.250 mg kg^−1^ of Zn, Mn, Cu, Fe, I, and Se, respectively; AACM70 or AACM-EZ70: 42, 49, 6, 28, 0.700, and 0.175 mg kg^−1^ of Zn, Mn, Cu, Fe, I, and Se, respectively; AACM40 or AACM-EZ40: 24, 28, 3, 16, 0.400, and 0.10 mg kg^−1^ of Zn, Mn, Cu, Fe, I, and Se, respectively. Data were analyzed by Dunnett’s test (*p* < 0.05); ° Differs from IM treatment; * Differs from IM-EZ treatment.

**Figure 5 animals-13-03709-f005:**
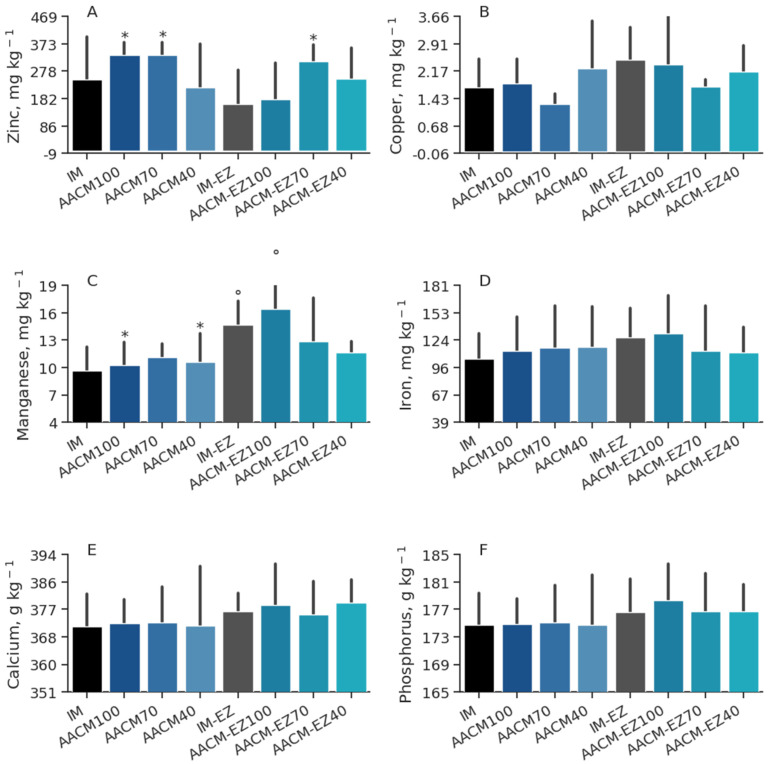
Zinc (**A**), copper (**B**), manganese (**C**), iron (**D**), calcium (**E**), and phosphorus (**F**) deposition in tibia of laying hens from 68 to 90 weeks fed inorganic minerals (IM) and amino acid complex minerals (AACM) supplemented with or without the use of phytase (EZ). IM or IM-EZ: 60, 70, 8, 40, 1.0, and 0.250 mg kg^−1^ of Zn, Mn, Cu, Fe, I, and Se, respectively; AACM100 or AACM-EZ100: 60, 70, 8, 40, 1.0, and 0.250 mg kg^−1^ of Zn, Mn, Cu, Fe, I, and Se, respectively; AACM70 or AACM-EZ70: 42, 49, 6, 28, 0.700, and 0.175 mg kg^−1^ of Zn, Mn, Cu, Fe, I, and Se, respectively; AACM40 or AACM-EZ40: 24, 28, 3, 16, 0.400, and 0.10 mg kg^−1^ of Zn, Mn, Cu, Fe, I, and Se, respectively. Data were analyzed by Dunnett’s test (*p* < 0.05); ^°^ Differs from IM treatment; * Differs from IM-EZ treatment.

**Figure 6 animals-13-03709-f006:**
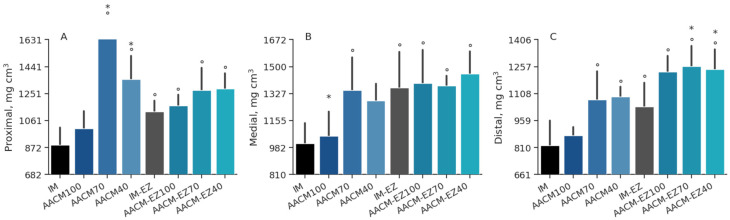
Densitometry of the proximal (**A**), medial (**B**), and distal (**C**) segments of the tibia in laying hens aged 68 to 90 weeks, fed with inorganic minerals (IM) and amino acid complex minerals (AACM), supplemented with or without the use of phytase (EZ). IM or IM-EZ: 60, 70, 8, 40, 1.0, and 0.250 mg kg^−1^ of Zn, Mn, Cu, Fe, I, and Se, respectively; AACM100 or AACM-EZ100: 60, 70, 8, 40, 1.0, and 0.250 mg kg^−1^ of Zn, Mn, Cu, Fe, I, and Se, respectively; AACM70 or AACM-EZ70: 42, 49, 6, 28, 0.700, and 0.175 mg kg^−1^ of Zn, Mn, Cu, Fe, I, and Se, respectively; AACM40 or AACM-EZ40: 24, 28, 3, 16, 0.400, and 0.10 mg kg^−1^ of Zn, Mn, Cu, Fe, I, and Se, respectively. Data were analyzed by Dunnett’s test (*p* < 0.05); ° Differs from IM treatment; * Differs from IM-EZ treatment.

**Figure 7 animals-13-03709-f007:**
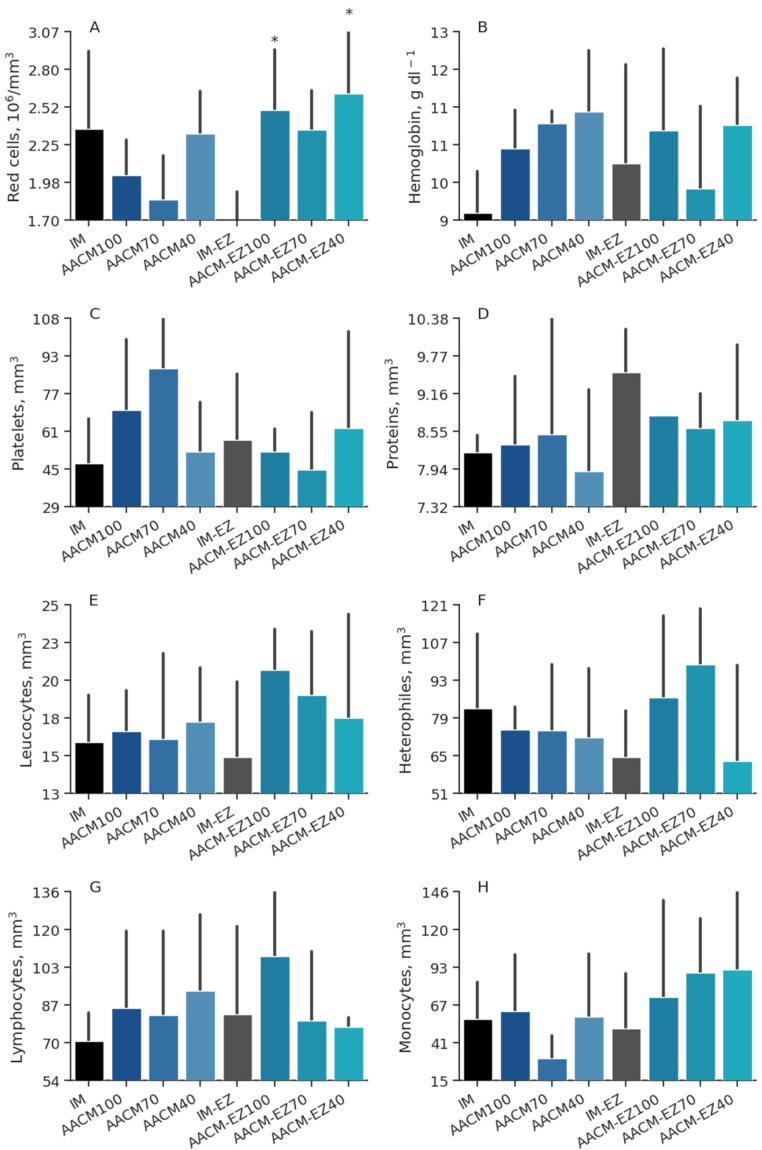
Red cells (**A**), hemoglobin (**B**), platelets (**C**), proteins (**D**), leucocytes (**E**), heterophiles (**F**), lymphocytes (**G**), and monocytes (**H**) blood profiles of laying hens from 68 to 90 weeks fed inorganic minerals (IM) and amino acid complex minerals (AACM) supplemented with or without the use of phytase (EZ).IM or IM-EZ: 60, 70, 8, 40, 0.250 mg kg^−1^ of Zn, Mn, Cu, Fe, I, and Se, respectively; AACM100 or AACM-EZ100: 60, 70, 8, 40, 0.250 mg kg^−1^ of Zn, Mn, Cu, Fe, I, and Se, respectively; AACM70 or AACM-EZ70: 42, 49, 6, 28, 0.175 mg kg^−1^ of Zn, Mn, Cu, Fe, I, and Se, respectively; AACM40 or AACM-EZ40: 24, 28, 3, 16, 0.100 mg kg^−1^ of Zn, Mn, Cu, Fe, I, and Se, respectively. Data were analyzed by Dunnett’s test (*p* < 0.05); * Differs from IM-EZ treatment.

**Figure 8 animals-13-03709-f008:**
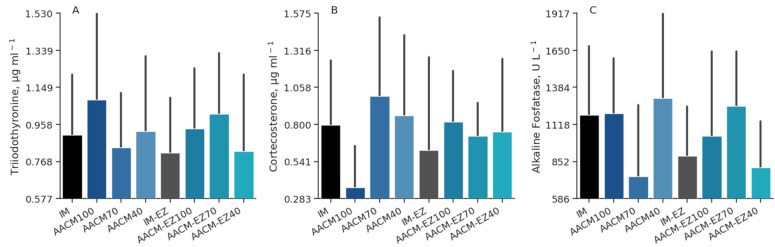
Triiodothyronine (**A**), corticosterone (**B**), and alkaline fosfatase (**C**) profiles of laying hens from 68 to 90 weeks fed inorganic minerals (IM) and amino acid complex minerals (AACM) supplemented with or without the use of phytase (EZ). IM or IM-EZ: 60, 70, 8, 40, 0.250 mg kg^−1^ of Zn, Mn, Cu, Fe, I, and Se, respectively; AACM100 or AACM-EZ100: 60, 70, 8, 40, 0.250 mg kg^−1^ of Zn, Mn, Cu, Fe, I, and Se, respectively; AACM70 or AACM-EZ70: 42, 49, 6, 28, 0.175 mg kg^−1^ of Zn, Mn, Cu, Fe, I, and Se, respectively; AACM40 or AACM-EZ40: 24, 28, 3, 16, 0.100 mg kg^−1^ of Zn, Mn, Cu, Fe, I, and Se, respectively. Data were analyzed by Dunnett’s test (*p* < 0.05).

**Table 1 animals-13-03709-t001:** Composition of experimental diets.

Ingredients, %	Without Phytase	With Phytase
Corn	58.756	58.756
Soybean meal, 46% CP	23.809	23.809
Soy oil	2.763	2.763
Calcitic limestone	10.538	10.654
Dicalcium phosphate	1.054	0.000
Meat and bone meal, 44%	1.580	1.580
Sodium bicarbonate	0.050	0.050
Salt	0.345	0.345
DL-Methionine, 99%	0.234	0.234
L-Lysine-HCl	0.140	0.140
L-Threonine	0.068	0.068
Adsorbent ^1^	0.100	0.100
Probiotic ^2^	0.050	0.050
Phytase ^3^	0.000	0.006
Vitamin Premix ^4^	0.100	0.100
Mineral Premix ^5^	0.285	0.285
Inert ^6^	0.121	1.054
Total	100.00	100.00
Nutritional composition		
Metabolizable energy, kcal kg^−1^	2820	2820
Dry matter ^7^, %	90.02	90.05
Crude protein, %	16.40	16.40
Crude protein ^7^, %	16.25	16.21
Ash, %	15.44	15.05
Digestible Methionine, %	0.45	0.45
Digestible Methionine + Cystine, %	0.68	0.68
Digestible Lysine, %	0.86	0.86
Digestible Threonine, %	0.61	0.61
Digestible Tryptophan, %	0.17	0.17
Calcium, %	4.50	4.50
Calcium ^7^, %	4.57	4.49
Total Phophorus ^7^, %	0.65	0.48
Available Phosphorus, %	0.37	0.37
Sodium, %	0.18	0.18
Chlorine, %	0.27	0.27
Potassium, %	0.60	0.60
Crude Fat, %	5.49	5.49

Guaranteed per kilogram of product: ^1^ Guaranteed levels: Bentonite, 666 g; Beta-glucans, 54 g; Mananoligossacaride, 59.4 g; Bio-atives Fitogenics, 16.5 g; ^2^ Guaranteed levels: *Bacillus licheniformis* (mín) > 16 × 10^10^ UFC g^−1^; ^3^ QUANTUM BLUE^®^—ABvista; Guaranteed levels: Phytase (min), 10,000 FTU kg^−1^; ^4^ Guaranteed levels (kg of product): Vitamin A, 8000 IU; Vitamin D3, 2000 IU; Vitamin E, 10,000 IU; Vitamin K3, 2000 mg; Vitamin B1, 1000 mg; Vitamin B2, 4000 mg; Vitamin B6, 2500 mg; Vitamin B12, 11,000 mg; Niacin, 25 g; Calcium pantothenate, 10 g; Folic acid, 550 mg; Biotin, 50 mg; ^5^ The composition is presented in Table 2; ^6^ Sand; ^7^ Analyzed Values.

**Table 2 animals-13-03709-t002:** Concentration of zinc (Zn), manganese (Mn), iron (Fe), copper (Cu), and selenium (Se) in calculated and analyzed diets, experimental premixes, and water.

	Calculated				
DIETS *	Zn	Mn	Fe	Cu	Se
	(mg kg^−1^)				
IM	60	70	40	8	0.250
AACM-100	60	70	40	8	0.250
AACM-70	42	49	28	5.6	0.175
AACM-40	24	28	16	3.2	0.100
	Diets analyzed *				
IM	71.6	73.3	288.9	11.1	0.320
AACM-100	73.4	76.3	265.0	10.6	0.313
AACM-70	58.3	53.3	249.5	7.95	0.286
AACM-40	39.3	36.4	218.4	4.65	0.220
IM-EZ	73.0	71.7	289.5	9.82	0.318
AACM-EZ-100	77.3	77.5	307.0	11.45	0.329
AACM-EZ-70	52.6	54.2	267.9	7.15	0.282
AACM-EZ-40	39.8	39.2	239.3	5.2	0.240
	Premixes analyzed *				
IM	61.8	68.0	46.6	7.4	0.265
AACM-100	62.1	71.1	44.4	7.6	0.255
AACM-70	42.3	48.2	33.6	5.5	0.180
AACM-40	26.8	27.4	26.6	3.0	0.080
WATER *	0.18	1.00	0.00	0.03	0.085

* Obtained by an inductively coupled plasma source. AACM: amino acid-complexed minerals; IM: Inorganic Minerals: Phytase enzyme (EZ).

**Table 3 animals-13-03709-t003:** Performance of laying hens from 68 to 90 weeks, supplemented with amino acid mineral complexes (AACM), with or without the use of phytase.

Enzyme	Level	EW	EO	EM	ADFI	FCR
		(g)	(%)	(g bird^−1^)		(kg kg^−1^)
AACM	-	64.95	92.53	60.09	103.21	1.717
AACM + Phytase	-	64.73	92.83	59.98	103.43	1.715
-	100	64.68	92.16	59.61	103.02	1.73
-	70	64.88	92.97	60.41	103.65	1.711
-	40	64.97	92.90	60.10	103.27	1.706
*p*-value						
Enzyme		0.581	0.583	0.887	0.367	0.828
Level		0.832	0.416	0.297	0.107	0.134
Enzyme × Level		0.363	0.023	0.101	0.041	0.030
SEM		0.192	0.29	0.222	0.12	0.007
		Interaction				
Variables		Levels				
Egg output		100		70		40
AACM		92.06 ^Aa^		91.83 ^Ba^		93.68 ^Aa^
AACM + Phytase		92.26 ^Aa^		94.11 ^Aa^		92.12 ^Aa^
ADFI						
AACM		102.64 ^Bb^		103.89 ^Aa^		103.16 ^Aab^
AACM + Phytase		103.52 ^Aa^		103.40 ^Aa^		103.37 ^Aa^
FCR						
AACM		1.721 ^Aa^		1.730 ^Aa^		1.703 ^Aa^
AACM + Phytase		1.759 ^Aa^		1.688 ^Ab^		1.710 ^Aab^

^Aa, Bb^ means followed by the same uppercase letters in the column, and lowercase letters in the row are considered similar by the Tukey test (*p* < 0.05); EW = egg weight; EO = egg output; EM = egg mass; ADFI = average daily feed intake; FCR = feed conversion ration; SEM: standard error of the mean; AACM-100 or AACM-EZ-100: 60, 70, 8, 40, 1.0, and 0.250 mg kg^−1^ of Zn, Mn, Cu, Fe, I, and Se, respectively; AACM-70 or AACM-EZ70: 42, 49, 6, 28, 0.700, and 0.175 mg kg^−1^ of Zn, Mn, Cu, Fe, I, and Se, respectively; AACM-40 or AACM-EZ-40: 24, 28, 3, 16, 0.400, and 0.100 mg kg^−1^ of Zn, Mn, Cu, Fe, I, and Se, respectively.

**Table 4 animals-13-03709-t004:** Quality of eggs of laying hens from 68 to 90 weeks, supplemented with amino acid mineral complexes (AACM), with or without the use of phytase.

Enzyme	Level	EW	YW	SW	AW	AH	ST	YC	HU
		(g)				(mm)		(score)	
AACM	-	64.27	17.02 ^B^	5.78	41.96	7.19	0.441	6.16 ^B^	82.98
AACM + Phytase	-	64.27	17.19 ^A^	5.74	41.93	7.20	0.435	6.37 ^A^	83.27
-	100	63.99	17.05	5.73	41.56	7.14	0.441	6.36 ^A^	82.72
-	70	64.31	17.13	5.74	41.99	7.25	0.435	6.30 ^AB^	83.57
-	40	64.48	17.14	5.79	41.86	7.12	0.438	6.16 ^B^	83.08
*p*-value									
Enzyme		0.987	0.01	0.497	0.88	0.814	0.111	< 0.01	0.504
Levels		0.631	0.559	0.718	0.899	0.148	0.453	0.054	0.274
Enzyme × Levels		0.678	0.001	0.155	0.679	0.576	0.136	0.336	0.601
SEM		0.168	0.034	0.023	0.132	0.026	0.002	0.034	0.211
		Interaction							
Variables		Levels							
Yolk weight, g		100			70			40	
AACM		16.908 ^Bb^			17.229 ^Aa^			16.945 ^Bb^	
AACM + Phytase		17.177 ^Aab^			17.051 ^Ab^			17.311 ^Aa^	

^A,a,B,b^ means followed by the same uppercase letters in the column, and lowercase letters in the row are considered similar by the Tukey test (*p* < 0.05); EW = egg weight; YW = yolk weight; SW = shell weight; AW = albumen weight; AH = albumen height; ST = Shell thickness; YC = Color of the yolk; HU = Haugh unit; SEM = standard error of the mean; AACM-100 or AACM-EZ-100: 60, 70, 8, 40, 1.0, and 0.250 mg kg^−1^ of Zn, Mn, Cu, Fe, I, and Se, respectively; AACM-70 or AACM-EZ70: 42, 49, 6, 28, 0.700, and 0.175 mg kg^−1^ of Zn, Mn, Cu, Fe, I, and Se, respectively; AACM-40 or AACM-EZ-40: 24, 28, 3, 16, 0.400, and 0.100 mg kg^−1^ of Zn, Mn, Cu, Fe, I, and Se, respectively.

**Table 5 animals-13-03709-t005:** Organ weight and intestine length of laying hens from 68 to 90 weeks, supplemented with amino acid mineral complexes (AACM), with or without the use of phytase.

Enzyme	Level	Liver	Spleen	Pancreas	Intestine	Oviduct	Intestine
		(g)					(m)
AACM	-	41.05	1.38	3.52 ^A^	63.66	69.11	1.37
AACM + Phytase	-	41.02	1.37	3.14 ^B^	64.62	65.88	1.37
-	100	40.01	1.33	3.24	61.86	66.70	1.31
-	70	42.98	1.48	3.58	65.31	68.64	1.41
-	40	39.98	1.30	3.17	65.25	66.45	1.36
*p*-value							
Enzyme		0.979	0.991	0.024	0.746	0.064	0.989
Level		0.225	0.269	0.098	0.405	0.452	0.234
Enzyme × Level		0.990	0.465	0.533	0.401	0.575	0.723
SEM		0.793	0.047	0.083	1.184	0.889	0.015

^A, B^ Means followed by the same uppercase letters in the column is considered similar by the Tukey test (*p* < 0.05); SEM = standard error of the mean; AACM-100 or AACM-EZ-100: 60, 70, 8, 40, 1.0, and 0.250 mg kg^−1^ of Zn, Mn, Cu, Fe, I, and Se, respectively; AACM-70 or AACM-EZ70: 42, 49, 6, 28, 0.700, and 0.175 mg kg^−1^ of Zn, Mn, Cu, Fe, I, and Se, respectively; AACM-40 or AACM-EZ-40: 24, 28, 3, 16, 0.400, and 0.100 mg kg^−1^ of Zn, Mn, Cu, Fe, I, and Se, respectively.

**Table 6 animals-13-03709-t006:** Trace mineral deposition in the tibia and egg yolk of laying hens from 68 to 90 weeks, supplemented with amino acid mineral complexes (AACM), with or without the use of phytase.

		Yolk						
Enzyme	Level	Zinc	Copper	Manganese	Iron	Calcium	Phosphorus	Selenium
				(mg kg^−1^)		(g kg^−1^)		(mg kg^−1^)
AACM	-	78.71 ^B^	3.92	2.63	136.75	3.02	12.01	0.35
AACM + Phyase	-	84.03 ^A^	3.41	2.66	125.54	3.01	12.12	0.34
-	100	82.49	3.86	2.82	137.9	2.96	12.11	0.38 ^A^
-	70	80.34	3.68	2.55	129.91	3.04	11.91	0.35 ^B^
-	40	81.6	3.47	2.58	126.16	3.04	12.16	0.31 ^C^
*p*-value								
Enzyme		0.003	0.464	0.837	0.542	0.778	0.649	0.358
Levels		0.746	0.571	0.678	0.903	0.541	0.705	<0.001
Enzyme × Level		0.074	0.882	0.135	0.674	0.362	0.584	0.789
SEM		2.481	0.203	0.186	12.411	0.025	0.247	0.007
		Tibia						
AACM	-	272.5	1.79	10.89 ^B^	115.69	372.48	175.08	-
AACM+Phyase	-	248.44	2.08	13.79 ^A^	119.19	377.58	177.48	-
-	100	237.83	2.08	13.51	122.54	375.42	176.8	-
-	70	304.17	1.53	12.25	115.07	374.18	176.06	-
-	40	237.21	2.2	11.38	114.36	375.7	176.03	-
*p*-value								
Enzyme		0.377	0.225	0.011	0.738	0.139	0.137	-
Levels		0.164	0.267	0.433	0.754	0.940	0.896	-
Enzyme × Level		0.152	0.477	0.139	0.682	0.853	0.874	-
SEM		17.26	0.14	0.55	5.54	1.66	0.77	-

^A, B, C^ means followed by the same uppercase letters in the column is considered similar by the Tukey test (*p* < 0.05); SEM = standard error of the mean. AACM-100 or AACM-EZ-100: 60, 70, 8, 40, 1.0, and 0.250 mg kg^−1^ of Zn, Mn, Cu, Fe, I, and Se, respectively; AACM-70 or AACM-EZ70: 42, 49, 6, 28, 0.700, and 0.175 mg kg^−1^ of Zn, Mn, Cu, Fe, I, and Se, respectively; AACM-40 or AACM-EZ-40: 24, 28, 3, 16, 0.400, and 0.100 mg kg^−1^ of Zn, Mn, Cu, Fe, I, and Se, respectively.

**Table 7 animals-13-03709-t007:** Bone densitometry of laying hens from 68 to 90 weeks, supplemented with amino acid mineral complexes (AACM), with or without the use of phytase.

Treatment	Level	Proximal *	Medial *	Distal *
AACM	-	1324	1232	1031
AACM + Phyase	-	1158	1374	1195
-	100	1183	1249	1057
-	70	1298	1269	1092
-	40	1244	1382	1174
	*p*-value			
Enzyme		0.042	0.015	0.001
Level		0.485	0.178	0.160
Enzyme × Level		0.019	0.001	0.038
SEM		45.25	37.29	28.83
	Interaction			
Variables	Levels			
Proximal		100	70	40
AACM		1107 ^Ab^	1509 ^Aa^	1359 ^Aab^
AACM + Phytase		1259 ^Aa^	1088 ^Ba^	1128 ^Aa^
Medial				
AACM		1062 ^Bb^	1359 ^Aa^	1300 ^Aa^
AACM+Phytase		1482 ^Aa^	1196 ^Ab^	1464 ^Aa^
Distal				
AACM		913 ^Bb^	1080 ^Aab^	1099 ^Aa^
AACM + Phytase		1236 ^Aa^	1107 ^Aa^	1250 ^Aa^

^Aa,Bb^ means followed by the same uppercase letters in the column and lowercase letters in the row are considered similar by the Tukey test (*p* < 0.05); SEM = standard error of the mean; AACM-100 or AACM-EZ-100: 60, 70, 8, 40, 1.0, and 0.250 mg kg^−1^ of Zn, Mn, Cu, Fe, I, and Se, respectively; AACM-70 or AACM-EZ70: 42, 49, 6, 28, 0.700, and 0.175 mg kg^−1^ of Zn, Mn, Cu, Fe, I, and Se, respectively; AACM-40 or AACM-EZ-40: 24, 28, 3, 16, 0.400, and 0.100 mg kg^−1^ of Zn, Mn, Cu, Fe, I, and Se, respectively. Mg/cm^3^.

**Table 8 animals-13-03709-t008:** Hematological, hormone, and alkaline phosphatase activity of laying hens from 68 to 90 weeks, supplemented with amino acid mineral complexes (AACM), with or without the use of phytase.

Enzyme	Level	Red Cells	Hematocrit	Hemoglobin	TP	Platelets *	Leukocytes *	Heterophiles **	Lymphocytes **	Monocytes *	T3	Corticosterone	AP
		(10^6^/mm^3^)	(%)	(g dL^−1^)		(mm^3^)					μg mL^−1^		U L^−1^
AACM	-	2.06 ^B^	32.66	11.00	8.17	7.00	16.97	7.39	8.69	5.11	0.95	1.01	1513
AACM + Phytase	-	2.48 ^A^	31.58	10.59	8.70	5.33	19.31	8.49	8.83	8.48	0.92	0.76	1281
-	100	2.26	32.00	10.76	8.45	6.13	18.90	8.09	9.68	6.82	1.01	0.59	2762
-	70	2.06	31.00	10.42	8.55	6.63	17.84	8.69	8.11	6.01	0.93	1.20	3075
-	40	2.43	33.37	11.21	8.25	5.75	17.68	6.83	8.50	7.57	0.87	0.80	3046
*p*-value													
Enzyme		0.010	0.485	0.419	0.550	0.163	0.210	0.390	0.684	0.191	0.836	0.569	0.575
Level		0.162	0.457	0.456	0.940	0.824	0.835	0.342	0.479	0.947	0.519	0.120	0.787
Enzyme × Level		0.845	0.491	0.389	0.910	0.202	0.696	0.447	0.558	0.446	0.414	0.084	0.225
SEM		0.096	0.736	0.248	0.280	5.856	0.733	4.972	0.505	0.797	0.051	0.012	2.0

^A,B^ means followed by the same uppercase letters in the column is considered similar by the Tukey test (*p* < 0.05); TP = Total protein; T3 = triiodothyronine; AP = alkaline phosphatase; SEM = standard error of the mean; AACM-100 or AACM-EZ-100: 60, 70, 8, 40, 0.250 mg kg^−1^ of Zn, Mn, Cu, Fe, I, and Se, respectively; AACM-70 or AACM-EZ-70: 42, 49, 6, 28, 0.175 mg kg^−1^ of Zn, Mn, Cu, Fe, I, and Se, respectively; AACM-40 or AACM-EZ-40: 24, 28, 3, 16, 0.100 mg kg^−1^ of Zn, Mn, Cu, Fe, I, and Se, respectively. Data were analyzed by the Tukey test (*p* < 0.05). * 10^3^; ** 10^4^.

## Data Availability

Data presented in this study are available upon request from the corresponding author.

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
