# Peer review of "The Impact of Phytase and Different Levels of Supplemental Amino Acid Complexed Minerals in Diets of Older Laying Hens"

_animals, 2023, doi:10.3390/ani13233709_

Round 1

Reviewer 1 Report

Comments and Suggestions for Authors

General:

This manuscript studied the impact of phytase and different levels of supplemental amino acid complexed minerals in diets of older laying hens. The resuts showed that the optimal amino acid-complexed minerals to better performance and egg quality with enzyme phytase. This article owned practical applied value, which provided regulation to direct manufacture and it is recommended to revise it before publication.

Specifics:

1. Research has shown that the addition of phytase promotes the absorption of calcium and phosphorus. What is the relationship between phytase addition and the metabolism or absorption of mineral elements such as zinc, copper, and manganese? This article does not provide sufficient explanation.

2. Why are there only 5 treatment groups in the tables of the results section, and why not list IM or IM-EZ group?

3. In the section of Experimental design and diets, it should clearly describe how many chickens each replicate contains.

4. In Table 1, why is dicalcium phosphate added in the group without phytase, but not in the phytase group? In adittion, the source and concentration of phytase product should be disclosed in Materials and Methods.

5. In 2.8, for blood biochemical analysis, 4 chickens were collected per replicate or 4 chickens were collected per treatment group. If 4 chickens were collected per treatment group, the sample size would be too small.

6. The organ weight in the material method should be adjusted forward and consistent with the order of results displayed. Replacing organ weight with the ratio of organ weight to body weight (organ index) is more accurate.

7. The same results do not need to be displayed repeatedly, for example Figure 1B and Figure 2A.

8. Error in ordinate data in Figure 4G.

9. The standard deviation of the data in Figure 7 and Figure 8 is larger than the data, such data is unconvincing.

10. In the Discussion, please explain why the requirements for Zn, Mn, Cu, Fe, Se, and I almost doubled when phytase was added. Meanwhile, why is the addition of AACM at a ratio of 100% not as good as the addition of 70% for some results? Should the absorption rate be reduced or antagonistic? Relevant explanations should be provided.

11. In the Note of Table 1.6x1010 UFC/g should be modified to 1.6x1010 UFC/g.

Author Response

General:

This manuscript studied the impact of phytase and different levels of supplemental amino acid complexed minerals in diets of older laying hens. The resuts showed that the optimal amino acid-complexed minerals to better performance and egg quality with enzyme phytase. This article owned practical applied value, which provided regulation to direct manufacture and it is recommended to revise it before publication.

Specifics:

  1. Research has shown that the addition of phytase promotes the absorption of calcium and phosphorus. What is the relationship between phytase addition and the metabolism or absorption of mineral elements such as zinc, copper, and manganese? This article does not provide sufficient explanation.

- Thank you for your valuable feedback. We have revised the manuscript, and the only table that explores the calcium and phosphorus results is Table 6, which displays the results of tibia and egg yolk deposition. However, neither result showed any significant statistical differences. That is why we did not explain these relationships in the manuscript.

  1. Why are there only 5 treatment groups in the tables of the results section, and why not list IM or IM-EZ group?

- Thank you for your comment. The experiment was conducted using a completely randomized experimental design with a 2×3+2 factorial arrangement. Three different tests were carried out. The first one compared AACM with and without phytase, totaling 6 treatments; the Tukey test was employed for this comparison. In the second test, all treatments were compared versus IM using the Dunnett test, and in the third test, all treatments were compared versus IM-EZ, also using the Dunnett test. It's important to note that five different P-values were generated, three for factorial, and 2 for Dunnet test. We attempted to present them in a single table in various ways. However, we have concluded that the design of all the tables is very confusing for readers. Presenting the results of the Dunnett test through figures enhances comprehension. 

  1. In the section of Experimental design and diets, it should clearly describe how many chickens each replicate contains.

- It has been described in the material and methods

  1. In Table 1, why is dicalcium phosphate added in the group without phytase, but not in the phytase group? In adittion, the source and concentration of phytase product should be disclosed in Materials and Methods.

- We appreciate your comment. The diets were formulated to achieve the available phosphorus. With the phytase treatment, the inclusion of 0.006 g/100 kg of the enzyme was sufficient to reach a quantity of 0.37 g/kg of available phosphorus in the diets. In the treatment without the enzyme, it was necessary to add 1.054 kg of dicalcium phosphate to achieve the same 0.37 g of available phosphorus. That was the reason why dicalcium phosphate was added in the group without phytase. According to the manufacturer, the contribution of available phosphorus is 3.250% with the inclusion of 60 g/ton. The source of phytase has been shown in the material and methods.

  1. In 2.8, for blood biochemical analysis, 4 chickens were collected per replicate or 4 chickens were collected per treatment group. If 4 chickens were collected per treatment group, the sample size would be too small.

-Thank you for your comment. The actual number of chickens was 1 bird per replication, totaling 8 birds per treatment. We have corrected this in the text.

  1. The organ weight in the material method should be adjusted forward and consistent with the order of results displayed. Replacing organ weight with the ratio of organ weight to body weight (organ index) is more accurate.

- The organ weight has been moved to the material and methods section, following the order of the Results.

  1. The same results do not need to be displayed repeatedly, for example Figure 1B and Figure 2A.

- Thank you for your comment, and I understand your perspective. However, these are two different results. In the performance data, the mean is calculated across all experimental periods. The egg weight for egg quality represents the mean of the last 3 days of each period. Showing only the performance egg weight could confuse the reader because the other variables would not be coherent with it.

  1. Error in ordinate data in Figure 4G.

- We have corrected the order of the variable in figure 4 and figure 5.

  1. The standard deviation of the data in Figure 7 and Figure 8 is larger than the data, such data is unconvincing.

- We appreciate your observation. Blood profile results in poultry often exhibit significant variability, as indicated in existing literature. A wide range of confidence intervals has been shown for these variables. In our study, we used the SEM to assess the precision of our results, the obtained SEM values indicate that our results fall within an acceptable range, given the nature of the variables being studied. This variability is a common feature in poultry studies, influenced by factors such as individual differences, environmental conditions, and others.

  1. In the Discussion, please explain why the requirements for Zn, Mn, Cu, Fe, Se, and I almost doubled when phytase was added. Meanwhile, why is the addition of AACM at a ratio of 100% not as good as the addition of 70% for some results? Should the absorption rate be reduced or antagonistic? Relevant explanations should be provided.

- Thank you for your comment, we have provide explanation in the discussion section.

  1. In the Note of Table 1.6x1010 UFC/g should be modified to 1.6x1010 UFC/g.

- Thank you for your comment, we have fixed it.

Reviewer 2 Report

Comments and Suggestions for Authors

The work is innovative as it seeks to present an ideal relationship between the supply of minerals in a complexed form or not and the impact of adding phytase depending on this relationship.

It is not clear in the description of the treatments, how the researchers met 0.37 of available phosphorus with the total removal of dicalcium phosphate, wouldn't it be better to put the real value and in the results justify that this is possible due to the high activity and efficiency of the enzyme ? Or describe the enzyme matrix.

Describe the meaning of the diet abbreviations in table 2. Every table must be self-explanatory.

Egg output was very high (92%), considering that they were animals aged 68 to 90 weeks.

Overall it is a very good and complete work.

Author Response

It is not clear in the description of the treatments, how the researchers met 0.37 of available phosphorus with the total removal of dicalcium phosphate, wouldn't it be better to put the real value and in the results justify that this is possible due to the high activity and efficiency of the enzyme ? Or describe the enzyme matrix.

-

  • We appreciate your comment. The diets were formulated to achieve the available phosphorus. With the phytase treatment, the inclusion of 0.006 g/100 kg of the enzyme was sufficient to reach a quantity of 0.37 g/kg of available phosphorus in the diets. In the treatment without the enzyme, it was necessary to add 1.054 kg of dicalcium phosphate to achieve the same 0.37 g of available phosphorus. That was the reason why dicalcium phosphate was added in the group without phytase. According to the manufacturer, the contribution of available phosphorus is 3.250% with the inclusion of 60 g/ton. 

Describe the meaning of the diet abbreviations in table 2. Every table must be self-explanatory.

- Thank you for your comment, we have fixed the missed meaning of abbreviations in the text

Egg output was very high (92%), considering that they were animals aged 68 to 90 weeks.

- Thank you for your comment. We appreciate your observation regarding the high egg output (92%) in animals aged 68 to 90 weeks. We understand your concern. However, these results were accurately obtained in our study. The high egg output observed could be attributed to the careful bird management provided during the experimental period

Reviewer 3 Report

Comments and Suggestions for Authors

I suggest to rearrange the methodology of the study: present the experimental design and the samples collected, and afterwards describe the methods used for analysis. Please see the comments in the attached file.

Author Response

Thank you for your time in reviewing our manuscript. We have agreed with all your comments and have followed your recommendations in the text. The answers to your questions are attached on this page.

Round 2

Reviewer 1 Report

Comments and Suggestions for Authors

There are currently no other opinions available.

Author Response

It appears that there are no comments from Reviewer 2.